# Measuring the continuation of antidepressant exposure prior to, during, and after pregnancy: A scoping review protocol

Lauren S. Tailor[1,2]*, Hilary K. Brown[2,3]ಠ, Jessie Cunningham[1,4]‡, Simone N. Vigod[5,6]‡, Erzsébet Horváth-Puhó[7]‡, Sonia M. Grandi[1,2]ಠ*

1 Child Health Evaluative Sciences, The Hospital for Sick Children, Toronto, Canada, 2 Division Epidemiology, Dalla Lana School of Public Health, University of Toronto, Toronto, Canada, 3 Department of Health and Society, University of Toronto Scarborough, Toronto, Canada, 4 Health Sciences Library, The Hospital for Sick Children, Toronto, Ontario, Canada, 5 Department of Psychiatry, Women's College Hospital, University of Toronto, Toronto, Canada, 6 Department of Psychiatry, Temerty Faculty of Medicine, University of Toronto, Toronto, Ontario, Canada, 7 Department of Clinical Medicine, Department of Clinical Epidemiology, Aarhus University, Aarhus, Denmark,

ಠ These authors contributed equally to this work.
‡ These authors also contributed equally to this work.
* lauren.tailor@mail.utoronto.ca (LT); sonia.grandi@sickkids.ca (SG)

## Abstract

### Objective

The goal of this scoping review is to summarize how prior studies have measured antidepressant continuation/discontinuation prior to, during, and after pregnancy.

## Introduction

Pregnant individuals and healthcare providers are faced with difficult decisions about whether to continue antidepressants in pregnancy due to the historical exclusion of pregnant women from clinical trials and the lack of rigorous evidence to support these decisions. Prior studies examining the effects of pre- and perinatal antidepressant use on perinatal outcomes using observational data have been inconsistent, primarily due to the binary (ever/never) categorization of exposure, which may not accurately reflect real-world use.

## Inclusion criteria

The population for this review consists of pregnant individuals. The concept is the measurement of continued preconception, prenatal, and postpartum antidepressant use. We will include human studies (no restrictions on language or geographic location) with the following study designs: cohort studies, cross-sectional studies, case-control studies, and descriptive analyses or spontaneous reports that reference antidepressant use over time.

**Data availability statement:** No datasets were generated or analysed during the current study. All relevant data from this study will be made available upon study completion.

**Funding:** LT is funded by the Canadian Institutes of Health Research's Vanier Canada Graduate Scholarship and an Edwin S.H. Leong Centre for Healthy Children 2025 Studentship Award. She was funded by a Network for Improving Health Systems (NIH) award from the University of Toronto Leslie Dan Faculty of Pharmacy at the start of this work. The funders did not play any role in the study design, data collection, analysis, decision to publish, or preparation of the manuscript. https://vanier.gc.ca/en/home-accueil.html, https://leongcentre.utoronto.ca/studentship-awards, https://www.nihs.pharmacy.utoronto.ca/.

**Competing interests:** I have read the journal's policy and the authors of this manuscript have the following competing interests: SNV reports royalties from UpToDate Inc. for authorship of materials on depression and pregnancy. This does not alter our adherence to PLOS ONE policies on sharing data and materials.

## Methods

We will conduct a scoping review using the JBI (formerly Joanna Briggs Institute) manual for scoping reviews and the Preferred Reporting Items for Systematic Reviews and Meta-Analyses extension for Scoping Reviews Checklist (PRISMA-ScR). The search strategy will be performed using database-specific nomenclature in MEDLINE, EMBASE, PsycINFO, Cochrane, Web of Science, and Canada's Drug Agency Grey Matters Guide for grey literature, limiting the final search to publications since 2022 to include contemporary data. Two independent reviewers will 1) screen titles/abstracts/full-texts and 2) extract data. Findings will summarize measurements of antidepressant continuation during the perinatal period, categorizing studies based on the descriptions of timing, duration, adherence, and exposure ascertainment.

## Conclusion

This scoping review will establish the extent to which prior studies have been able to measure continued use to inform a clear definition of continued antidepressant exposure to be used in future studies. **Review registration:** Open Science Framework https://osf.io/2ewpq

---

## Introduction

Mental health disorders, such as depression and anxiety, are highly prevalent in pregnancy, with approximately 20–25% of pregnant individuals experiencing symptoms of depression or anxiety [1–4]. If untreated, depression and anxiety can negatively impact maternal well-being and offspring health, emphasizing the need for appropriate treatment, such as antidepressant therapy [1–4]. Antidepressants are among the most commonly prescribed medications in pregnancy, with a global prevalence ranging from 3–8% [5, 6]. However, prenatal antidepressant use has been associated with an elevated risk of certain adverse maternal and fetal health outcomes, such as gestational diabetes and preterm birth [7–13]. Consequently, pregnant individuals and healthcare providers are faced with difficult decisions about whether to continue antidepressants when trying to conceive, during pregnancy, and post-partum (i.e., during breastfeeding) based on factors such as disease severity, potential risks, and patient and provider experience and preference [14–17]. These decisions are complicated by uncertainty surrounding the effect of continuing antidepressant use throughout the perinatal period (i.e., continuing throughout pregnancy vs. discontinuing) on maternal and offspring health, limiting evidence-based clinical decision-making. Therefore, there is a need to examine the effect of continuing antidepressants in pregnancy on maternal and offspring outcomes.

While clinicians often rely on randomized trials as the gold standard for estimating causal effects, these studies typically exclude pregnant or breastfeeding individuals due to ethical concerns over maternal and fetal safety and feasibility issues (e.g., limited pool of eligible/willing participants, rare exposures/

outcomes, and challenges with long-term follow-up) [14-16, 18]. As such, healthcare providers and patients must frequently make treatment decisions based on post-marketing surveillance data, health registries, case reports, and observational studies using health administrative and claims databases [14, 19]. Importantly, prior observational studies commonly classify use of prenatal medications as "ever" vs. "never exposed" throughout pregnancy or by trimester, which does not accurately reflect real-world sustained use of medications and ignores important aspects, such as timing of initiation and discontinuation [15, 16, 20]. Considering the timing of medication initiation is crucial in studies examining medication use during pregnancy, as there are critical periods of fetal development when the fetus may be more vulnerable to the harmful effects of certain medications or exposures [21, 22]. For example, classifying antidepressant timing solely by trimester can lead to exposure misclassification, as it ignores sensitive time windows within each trimester; for instance, the embryonic heart undergoes major structural development between weeks 3–8 of gestation [23], so a general "first trimester" exposure categorization could obscure associations between antidepressant exposure during this time window and congenital heart defects [16]. While some studies report the duration of exposure (e.g., number of weeks or trimesters of use) or classify medication use based on dose (high, medium, low dose), there is a lack of standardization in how cumulative duration and dose-response are captured, and these approaches often do not adequately account for changes in medication exposure or dose over time [16]. As such, more granular definitions of real-world antidepressant exposure during pregnancy are needed, particularly since outcomes have periods of sensitivity to specific treatment exposures, meaning clinicians and patients may be likely to alter treatment regimens early in or before the perinatal period (e.g., they may decide to continue, modify, or discontinue therapy). Longitudinal measures of continuation or discontinuation of medication use that account for preconception patterns of antidepressant use are therefore necessary.

Research focused on the sustained use of antidepressants after pregnancy and into lactation is also limited [24], and improved methods of measuring the continuation of postpartum antidepressant use are needed. Moreover, the ascertainment of key characteristics of antidepressant use, such as dose, class, timing, duration, and indication for use, varies across studies, underscoring the need for a standardized approach. Establishing consistency in these measures is essential, as they can significantly influence clinicians' and patients' decisions on whether to continue or discontinue antidepressant treatment before, during, or after pregnancy, including during lactation (as this may impact the decision of formula vs. breastfeeding).

As such, this review aims to identify and summarize the extent to which prior studies of perinatal antidepressant exposure have accounted for or measured the continuation of antidepressant exposure prior to, during, and after pregnancy among pregnant individuals in all scoping review contexts (i.e., geographic, social, or cultural settings) [25]. A preliminary search of MEDLINE, Open Science Framework (OSF), the Cochrane Database of Systematic Reviews and *JBI Evidence Synthesis* was conducted to identify current or underway systematic reviews or scoping reviews on the topic of measuring continued antidepressant exposure throughout the perinatal period, and none were identified. Scoping review methodology is more appropriate here than a systematic review or meta-analysis as the intent is to clarify key concepts/definitions in the literature, and identify and analyze knowledge gaps, as opposed to forming a conclusion around a prevalence, risk factor or effect size of an intervention [26].

### Review question(s)

i. How have previous studies measured the continuation of antidepressant exposure a) prior to, b) during, and c) after pregnancy (i.e., binary exposures, sustained use, measurement of lactation)?

ii. What specific characteristics have previous studies measured concerning continued antidepressant exposure prior to, during, and after pregnancy (e.g., indication, dose, duration, timing)?

 

## Materials and methods

### Ethics approval

This review is part of LT's doctoral thesis, which has received ethics approval from the Research Ethics Boards of the University of Toronto (HPR-00055550-V0001.0000-O), Aarhus University (REB# 2016-051-000001/810), and the Hospital for Sick Children (REB# REB 1000080862).

### Eligibility criteria

### Participants

As described in [Table 1](), we will include studies examining human pregnant individuals who continue to use antidepressants throughout the perinatal period, defined as use during pregnancy as well prior to conception (within 12 months preconception) continuing into pregnancy and after pregnancy (continued use within 12 months post-delivery), with a focus on sustained use throughout pregnancy (i.e., not postpartum initiation of antidepressants). We will exclude studies that focus on non-human populations or only include individuals who are not pregnant, attempting to conceive, or have a

**Table 1. Eligibility Criteria.**

| Inclusion Criteria | Exclusion Criteria | Additional Notes |
|---|---|---|
| *Title/Abstract Screening*<br>• Peer-reviewed studies published from **2015** onward will be included in <u>title and abstract screening</u><br>*Full-Text Review*<br>• Peer-reviewed studies published from **2022** onward will be included in the <u>full-text review</u><br>**Population**<br>• Human individuals who were **assigned female sex at birth** and are of **reproductive age** (typically 15–49 years].<br>• Individuals were **pregnant** during the study period.<br>**Exposure**<br>• Use of **antidepressant medications**:<br>  ◦ **During pregnancy**<br>  ◦ **Prior to conception** [within 12 months preconception) and into pregnancy<br>  ◦ **Postpartum** (within 12 months post-delivery) from pregnancy<br>• Antidepressant use must be the **primary exposure** of interest<br>**Study Designs**<br>• Study design must be able to capture **sustained antidepressant use throughout pregnancy**.<br>• We will include the following study types that are able to measure individual-level longitudinal patterns of antidepressant use:<br>  ◦ **Cohort studies**<br>  ◦ **Case-control studies**<br>  ◦ **Cross-sectional studies** (e.g., those using retrospective recall across preconception, pregnancy, and postpartum periods)<br>  ◦ **Descriptive analyses** or **spontaneous reports** that reference antidepressant use over time | *Title/Abstract Screening*<br>• Peer-reviewed studies published prior to **2015** will be excluded in <u>title and abstract screening</u><br>*Full-Text Review*<br>• Peer-reviewed studies published prior to **2022** will be excluded in the <u>full-text review</u><br>**Population**<br>• Studies conducted in **non-human populations**<br>• Studies including only:<br>  ◦ Individuals who are **not pregnant**<br>  ◦ Individuals **attempting to conceive**<br>  ◦ Individuals with a history of **delivery more than 12 months prior**<br>**Exposure**<br>• Studies that examine **postpartum initiation** of antidepressants only (i.e., do not assess sustained use through pregnancy and into the postpartum period)<br>• Antidepressant use is **not the primary exposure** of interest<br>**Study Designs**<br>• Studies **lacking the ability to measure individual-level longitudinal patterns** of antidepressant use throughout the perinatal period across broad populations<br>• Studies with insufficient primary data or detail of perinatal antidepressant use for epidemiological analyses, including:<br>  ◦ **Conference abstracts**<br>  ◦ **Case reports/series**<br>  ◦ **Editorials**<br>  ◦ **Reviews**<br>  ◦ **Opinion pieces**<br>  ◦ **Qualitative studies**<br>• **Grey literature** that does not include **longitudinal, individual-level data** on antidepressant exposure | • The initial title and abstract screening will include studies from 2015 onward to capture a broad range of exposure measurement approaches and reflect evolving antidepressant use and methodological developments since 2015, but full-text screening will be limited to studies published from 2022 onward. This change was made to:<br>  ◦ Improve feasibility given the high volume of literature<br>  ◦ Focus on contemporary patterns of antidepressant use, reflecting changes in prescribing practices over the past decade<br>• The review will not impose restrictions based on sex, gender, or age in the search strategy<br>• This review will consider all human studies with no restrictions on language, geographic location, or social-cultural contexts (e.g., acute care).<br>• Inclusion will be based on relevance to perinatal antidepressant exposure in pregnant or previously pregnant individuals |

history of delivery more than 1 year prior. We will not impose other population restrictions (i.e., we will include individuals with comorbidities or co-existing conditions) [27]. Concerning sex and gender, this review will focus on perinatal antidepressant exposure, meaning only conceiving, pregnant, or previously pregnant individuals of reproductive age (assigned female sex at birth, generally 15–49 years [28, 29]) will inherently be included, although we will not limit the search by sex, gender, or age.

## Concept

The main concept of interest for this scoping review is how prior studies have measured and defined continued antidepressant use throughout the preconception, prenatal, and postpartum periods, for all antidepressant classes, particularly concerning continued use throughout the perinatal period and the categorization of continued antidepressant use (i.e., binary, categorical (i.e., ever, never, sometimes)). Since the focus of our concept is the measurement of continued antidepressant use during the perinatal period, we are also interested in specific factors that are measured in relation to antidepressant exposure, such as indication, dose, class, trimester/timing, and/or cumulative duration of use. Therefore, an additional element of our concept of interest will be the characteristics of antidepressant use.

## Context

This review will consider all human studies with no restrictions on language or geographic location. As described by Peters et al., the context of a scoping review includes details on location, social-cultural factors, or sex-based factors [27]. We will not restrict based on location or social-cultural contexts (e.g., acute care).

## Types of sources

As per Peters et al., reviews can limit study designs based on the research question and when the team believes certain designs are unlikely to contain relevant information [25]. Since we aim to summarize continued perinatal antidepressant exposure, we will include study designs that allow for the analysis of temporal assessment of exposure with longitudinal data prior to, during, or following a pregnancy. We will, therefore, limit our search to peer-reviewed studies that capture patterns or trends in exposure with the following study designs: cohort studies, cross-sectional studies (i.e., where women recall preconception, prenatal, and postpartum use), case-control studies, and descriptive analyses or spontaneous reports that reference use of medications over time. We will exclude studies that lack the ability to measure individual-level longitudinal patterns of antidepressant use throughout the perinatal period across broad populations, as well as those that have insufficient primary data or detail of use (e.g., conference abstracts, case reports/series, editorials, reviews, and opinion pieces). A search of the grey literature will be performed using the first five pages of Google, Google Scholar, and Canada's Drug Agency Grey Matters tool (formerly the Canadian Agency for Drugs and Technologies in Health (CADTH) Grey Matters Guide) Advisories and Warnings section to examine surveillance data from pharmaceutical or public health agencies [30]. As with peer-reviewed studies, included grey literature must have longitudinal data of individual-level antidepressant exposure to be included.

## Study design

The proposed scoping review will be conducted in accordance with the JBI methodology for scoping reviews [31] and the protocol will be reported according to the Preferred Reporting Items for Systematic Reviews and Meta-Analyses extension for protocols (PRISMA-ScR) [27, 32]. As noted by Munn et al. and previously described, scoping reviews are indicated when researchers seek to clarify key concepts/definitions in the literature and identify and analyze knowledge gaps, making the scoping review format the most appropriate for this research question [26]. This protocol has been registered in OSF (https://osf.io/2ewpq).

## Search strategy

The search strategy aims to locate published studies. A three-step search strategy will be used in this review as per the JBI Manual for Evidence Synthesis [33]. First, an initial limited search of MEDLINE (PubMed) will be conducted to identify articles on the topic. The text words contained in the titles and abstracts of relevant articles and the index terms used to describe the articles will be used to develop a full search strategy for Ovid MEDLINE(R) and Epub Ahead of Print, In-Process & Other Non-Indexed Citations and Daily; Ovid Embase Classic + Embase; Ovid APA PsycInfo; Ovid EBM Reviews Central Register of Clinical Trials; and Clarivate Web of Science Core Collection (see S1 File. Complete Literature Search Strategy). Grey literature sources will be identified using previously described methods. With guidance from an expert librarian (JC), the search strategy, including all identified keywords and index terms, will be adapted for each included database and/or information source. The reference lists of all included sources of evidence will be screened for additional studies.

While the initial abstract screening will include studies published from 2015 onward to capture a broad range of potentially relevant literature, the full-text review will be limited to studies published from 2022 to 2024. This decision was made to improve feasibility given the high volume of eligible studies and to ensure that the review reflects contemporary patterns of antidepressant use during pregnancy. Over the past two decades, prenatal antidepressant prescribing has shifted, with a decline in the use of older, less selective antidepressants and a rise in the use of more selective agents, such as selective serotonin reuptake inhibitors (SSRIs). Limiting the full-text review to more recent publications ensures a focus on research that aligns with current prescribing practices, safety profiles, and clinical guidelines, thereby increasing the relevance of our findings for contemporary clinical decision-making [34]. This timeframe limitation was chosen pragmatically to manage the volume of literature and ensure relevance to contemporary prescribing patterns, while still capturing a range of exposure measurement approaches. Non-English language articles will be translated when needed, and artificial intelligence software (e.g., Deep-L [35–37] and CUBBITT [37, 38] have been shown to be highly reliable, with CUBBIT exhibiting increased fluency) to translate non-English studies may be used in addition to human translators, as have been used in previous scoping reviews [39].

## Study selection

Following the search, all identified citations will be collated and uploaded into Covidence screening software (Veritas Health Innovation, Melbourne, Australia), and duplicates will be removed. All identified titles/abstracts will be screened independently by at least two independent reviewers based on the inclusion criteria for the review. We will conduct a pilot test of 50 titles/abstracts and 10 full-text articles prior to conducting the official screening to ensure that the eligibility criteria are sufficient and clear, with a goal of 75% agreement [25]. The references of included studies will be scanned to identify studies not identified in the original search. All studies identified in the title/abstract screening stage that meet the inclusion criteria will proceed to full-text review. At the full-text review stage, we will apply more detailed eligibility criteria and extract relevant data. Reasons for the exclusion of full-text papers that do not meet the inclusion criteria will be recorded and reported in the scoping review. Any conflicts will be resolved through discussion or by a third independent reviewer. Search results and the study inclusion process will be reported in full in the final scoping review and presented in a PRISMA flow diagram [40].

## Data extraction

Relevant information from the studies, including year of publication, country, design, objectives, study population, definition and measurement of continued pre- and perinatal antidepressant use, indication, method of ascertainment of indication, and outcomes examined, will be extracted by two independent reviewers using a data extraction tool developed by the reviewers. Any conflicts will be resolved through discussion or by a third independent reviewer. A risk of bias assessment

will not be conducted as per scoping review guidelines [25]. The draft data extraction tool (refer to Table 2) will be modified and revised as necessary during the data extraction process, and all modifications will be detailed in the full scoping review. If appropriate, the authors of included papers will be contacted to request missing or additional data, where required.

Since this study is a scoping review, all data underlying the findings will be derived from published, peer-reviewed articles or grey literature that are already available in the public domain. A complete list of included studies will be provided as a supplementary file with the final manuscript. The extracted data will also be made available as supplementary tables at the time of publication.

## Data analysis and presentation

We will use descriptive statistics to summarize the characteristics of included studies (e.g., the number and proportion of studies reporting specific medication classes/types). A summary of the characteristics of antidepressant use (i.e., dose, class, trimester/timing, and duration) will also be displayed in tables. To ascertain how continued antidepressant use has been captured, we will first report the number of studies that consider versus do not consider antidepressant continuation during pregnancy and then will categorize studies based on the robustness of their measurement of

**Table 2. Draft data extraction instrument[a.]**

| | | Study 1 | Study 2 | Study x |
|---|---|---|---|---|
| **Study Characteristics** | **Number** | | | |
| | **Title** | | | |
| | **Covidence #** | | | |
| | **DOI** | | | |
| | **Reviewer Initials** | | | |
| | **Covidence Tags** | | | |
| | **Authors** | | | |
| | **Pub Year** | | | |
| | **Journal Name/Grey Literature** | | | |
| | **Country (author affiliations)** | | | |
| | **Study Design Type**<br>1. Prospective Cohort Study<br>2. Retrospective Cohort Study<br>3. Case–Control Study<br>4. Cross-Sectional Study<br>5. Descriptive Analysis<br>6. Spontaneous Report<br>7. Nested Case Control (from a base cohort)<br>8. Randomized controlled trial<br>9. Other (describe what you mean by "other" in the comments) | | | |
| | **Study period (years)** | | | |
| | **Data Source Type**<br>1. Administrative claims/ registries<br>2. Electronic health records (EHR) (hospital data)<br>3. Pharmacy dispensing data only<br>4. Survey or self-report<br>5. Mixed (specify in comments)<br>Other (specify in comments) | | | |
| | **Additional Exposures** | | | |
| | **Outcome** | | | |
| **Population Characteristics** | **Brief Description of Inclusion Criteria** | | | |
| | **Brief Description of Exclusion Criteria** | | | |
| | **Sample size (per strata if applicable)** | | | |

*(Continued)*

**Table 2.** (Continued)

|  |  | Study 1 | Study 2 | Study x |
|---|---|---|---|---|
| **Antidepressant Exposure Characteristics** | **Was pre-pregnancy antidepressant use reported or considered?** (Y/N/Unclear) |  |  |  |
|  | **If yes, how was pre-pregnancy use accounted for?** <br> 1. Inclusion or exclusion criteria (e.g., restricted to those with use in pre-pregnancy) <br> 2. Within exposure definition (e.g., continuers vs. discontinuers vs. initiators) <br> 3. Adjusted for as a covariate <br> 4. Used as a stratification variable <br> 5. Mixed (specify) <br> 6. Other (specify) |  |  |  |
|  | **Measurement of exposure (Method for capturing exposure, e.g., codes, health records)** |  |  |  |
|  | **List of antidepressant classes and names studied** |  |  |  |
|  | **Were patterns of antidepressant exposure (e.g., sustained or continued use) described?** (Y/N/Unclear) |  |  |  |
|  | **If yes: Please describe how prenatal antidepressant continuation, discontinuation or other patterns of use were measured:** <br> 1. Categories or trajectories of continuation <br> 2. Time-varying exposure <br> 3. Prescription count/dose and timing to define continuation <br> 4. Duration or adherence measures <br> 5. Definition of discontinuers <br> 6. Episodic or interrupted use <br> 7. Dose tapering throughout pregnancy <br> 8. Switching of medications captured throughout pregnancy <br> 9. Continuous Timing of Exposure Measurement <br> 10. Other (describe) |  |  |  |
|  | **If yes: List the Number(s) of Descriptions Identified** |  |  |  |
|  | **If no or unclear: Please describe how prenatal antidepressant exposure was defined in the study** |  |  |  |
|  | **Was continuation of antidepressant use into the postpartum period reported or considered?** (Y/N/Unclear) |  |  |  |
|  | **If yes, how was post-partum use accounted for?** |  |  |  |
| **Reviewer comments** |  |  |  |  |

[a]Please note that this table has been transposed vertically for protocol presentation purposes. The final version will be completed in Excel using a horizontal format.

continued antidepressant use (considering factors such as timing, duration, adherence, exposure ascertainment, etc.). Study details will be presented as tables or figures (e.g., word heat maps, waffle charts) as described by Pollock et al [41] where appropriate. Tabular or diagrammatic data will be accompanied by a narrative summary to illustrate key findings.

## Scoping review timeline

As this is a scoping review of the available literature, no participant recruitment is involved. Title and abstract screening have been completed as of June 18, 2025. Full-text screening has been completed as of July 7, 2025. Data extraction is underway and expected to be completed by October 2025. We anticipate that results will be available by December 2025.

## Discussion

This protocol provides an overview of the methods for conducting a scoping review to summarize how prior studies have measured continued and sustained antidepressant exposure prior to, during, and after pregnancy. This review will identify key methodological strengths, limitations, and gaps in the literature, ultimately informing a clear definition of continued antidepressant exposure to be used in future studies. Establishing a clear and consistent definition of sustained chronic medication use, such as antidepressant use, in pregnancy is critical for improving comparability across studies, enhancing the validity of findings, and guiding clinical and policy decision-making regarding perinatal antidepressant use. The results from this review will be disseminated widely through conference presentations (e.g., The European Perinatal and Paediatric Epidemiology Conference) and at least one peer-reviewed journal publication.

## Supporting information

**S1 File. Complete Literature Search Strategy.**
(DOCX)

**S2 File. Checklist- PRISMA-P 2015 Checklist.**
(DOCX)

## Acknowledgments

This review will contribute to the doctoral thesis of LT.

## Author contributions

**Conceptualization:** Lauren Tailor, Hilary K. Brown, Simone N. Vigod, Sonia M. Grandi.

**Formal analysis:** Lauren Tailor.

**Methodology:** Lauren Tailor, Hilary K. Brown, Jessie Cunningham, Simone N. Vigod, Erzsébet Horváth-Puhó, Sonia M. Grandi.

**Supervision:** Hilary K. Brown, Simone N. Vigod, Sonia M. Grandi.

**Writing – original draft:** Lauren Tailor.

**Writing – review & editing:** Lauren Tailor, Hilary K. Brown, Jessie Cunningham, Simone N. Vigod, Erzsébet Horváth-Puhó, Sonia M. Grandi.

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
