## [Decision Letter · Decision Letter 0]

13 Aug 2025

PONE-D-25-32258Measuring the continuation of antidepressant exposure prior to, during, and after pregnancy: a scoping review protocolPLOS ONE

Dear Dr. Tailor,

Thank you for submitting your manuscript to PLOS ONE. After careful consideration, we feel that it has merit but does not fully meet PLOS ONE’s publication criteria as it currently stands. Therefore, we invite you to submit a revised version of the manuscript that addresses the points raised during the review process.

We look forward to receiving your revised manuscript.

Kind regards,

Xiaoqin Liu

Academic Editor

PLOS ONE

Journal Requirements: 

[I have read the journal's policy and the authors of this manuscript have the following competing interests: SNV reports royalties from UpToDate Inc. for authorship of materials on depression and pregnancy.].

Reviewers' comments:

Reviewer's Responses to Questions

**Comments to the Author**

1. Does the manuscript provide a valid rationale for the proposed study, with clearly identified and justified research questions?

Reviewer #1: Yes

Reviewer #2: Partly

2. Is the protocol technically sound and planned in a manner that will lead to a meaningful outcome and allow testing the stated hypotheses?

Reviewer #1: Yes

Reviewer #2: Partly

3. Is the methodology feasible and described in sufficient detail to allow the work to be replicable?

Reviewer #1: Yes

Reviewer #2: Yes

4. Have the authors described where all data underlying the findings will be made available when the study is complete?

Reviewer #1: Yes

Reviewer #2: No

5. Is the manuscript presented in an intelligible fashion and written in standard English?

Reviewer #1: Yes

Reviewer #2: Yes

6. Review Comments to the Author

You may also provide optional suggestions and comments to authors that they might find helpful in planning their study.

Reviewer #1: Dr. Tailor and colleagues have developed a well-structured and clearly articulated review protocol aimed at summarizing how antidepressant continuation before, during, and after pregnancy is addressed in human studies. The protocol is methodologically sound and thoughtfully prepared. I have only two minor comments and questions that the authors may wish to consider:

1. Table 1. Eligibility criteria: Under the inclusion criteria, the authors state: “Peer-reviewed studies published from 2022 onward will be included in the full-text review.” In contrast, under the exclusion criteria, they write: “Peer-reviewed studies published prior to 2022 will be excluded from the full-text review.” While both statements are technically consistent, presenting the same information in both sections creates redundancy and may be confusing. Moreover, exclusion from full-text review implies that earlier studies will not be screened at that stage, not that they meet the exclusion criteria per se. I suggest rephrasing this section for clarification.

2. The decision to limit the full-text review to studies published between 2022 and 2024 is under-standable, particularly given the high volume of eligible studies and the desire to reflect contempo-rary patterns of antidepressant use during pregnancy. However, to strengthen the rationale for this cutoff, the authors might consider conducting a consistency check between the title/abstract screening and full-text review stages within the 2022–2024 subset. Specifically, it would be in-formative to report on the proportion of studies that are included after full-text review but would have been missed if relying solely on title and abstract screening. This would help demonstrate the added value of the full-text review in refining study inclusion and provide empirical justification for the chosen approach.

Reviewer #2: Thank you for the opportunity to review this scoping review protocol. This scoping review aims to summarize how prior studies have measured antidepressant continuation/discontinuation prior to, during, and after pregnancy. While this is an important topic, I have several concerns that the authors might want to consider.

1. While the authors acknowledged that "timing of medication initiation is crucial in studies examining medication use during pregnancy" and provided examples of biological windows relevant to specific outcomes, the protocol did not attempt to classify studies by outcome type. This lack of focus led to the inclusion of a large number of studies, far exceeding the capacity of the research team to manage effectively. As a result, it seems infeasible to issue meaningful recommendations for future research based on the findings of this review. It also raised concerns about the pragmatic rationale behind including studies from 2015 based only on title and abstract screening, while including studies from 2022 based on full-text review.

2. The authors noted that "more granular definitions of real-world antidepressant exposure during pregnancy are needed, particularly since outcomes have periods of sensitivity to specific treatment exposures, meaning clinicians and patients may be likely to alter treatment regimens early in or before the perinatal period." However, the feasibility of such granular definitions largely depends on the type of data available. In many cases, particularly in studies relying on self-reported data, this level of detail is simply not attainable. Despite this, the authors did not include the type of data source among the extracted variables, which represents a significant oversight.

7. PLOS authors have the option to publish the peer review history of their article (what does this mean? ). If published, this will include your full peer review and any attached files.

**Do you want your identity to be public for this peer review?** For information about this choice, including consent withdrawal, please see our Privacy Policy .

Reviewer #1: No

Reviewer #2: No

---

## [Author Response · Author response to Decision Letter 1]

26 Sep 2025

Response to Editor and Reviewers’ Comments - PONE-D-25-32258

Dear Dr. Chenette and Dr. Liu,

We appreciate all the thoughtful comments from the Editor and reviewers. Please find our responses to each comment below, and the corresponding changes we have made in the revised manuscript. The reference to page numbers in the response reflects those from the tracked changed version of the manuscript.

Editor’s comments:

1. Please ensure that your manuscript meets PLOS ONE's style requirements, including those for file naming. The PLOS ONE style templates can be found at https://journals.plos.org/plosone/s/file?id=wjVg/PLOSOne_formatting_sample_main_body.pdf and https://journals.plos.org/plosone/s/file?id=ba62/PLOSOne_formatting_sample_title_authors_affiliations.pdf.

We thank the Editor for this note, and have revised the manuscript to reflect PLOS ONE’s Style requirements, including the naming of the supplementary documents.

2. We note that the grant information you provided in the ‘Funding Information’ and ‘Financial Disclosure’ sections. When you resubmit, please ensure that you provide the correct grant numbers for the awards you received for your study in the ‘Funding Information’ section.do not match.

We have revised the funding information and financial disclosure sections accordingly on page 2 of the manuscript and within the Financial Disclosure section, as follows: “LT is funded by a Vanier Canada Graduate Scholarship (Funding Reference Number: CGV-192654), Canadian Institutes of Health Research Michael Smith Foreign Study Supplements Award (Funding Reference Number: FSS-203589), and an Edwin S.H. Leong Centre for Healthy Children 2025 Studentship Award. She was funded by a Network for Improving Health Systems Trainee award from the University of Toronto Leslie Dan Faculty of Pharmacy and Dalla Lana School of Public Health at the start of this work.”

3. Thank you for stating the following in the Competing Interests section: [I have read the journal's policy and the authors of this manuscript have the following competing interests: SNV reports royalties from UpToDate Inc. for authorship of materials on depression and pregnancy.]. Please confirm that this does not alter your adherence to all PLOS ONE policies on sharing data and materials, by including the following statement: ""This does not alter our adherence to PLOS ONE policies on sharing data and materials.” (as detailed online in our guide for authors http://journals.plos.org/plosone/s/competing-interests). If there are restrictions on sharing of data and/or materials, please state these. Please note that we cannot proceed with consideration of your article until this information has been declared. Please include your updated Competing Interests statement in your cover letter; we will change the online submission form on your behalf.

We have revised the Competing Interests section to include the above statement, and have included this within the cover letter, as follows: “I have read the journal's policy, and the authors of this manuscript have the following competing interests: SNV reports royalties from UpToDate Inc. for authorship of materials on depression and pregnancy. This does not alter our adherence to PLOS ONE policies on sharing data and materials.”

We have moved the Ethics statement to appear only in the Methods section of the manuscript.

N/A

Reviewers’ comments:

1. Have the authors described where all data underlying the findings will be made available when the study is complete?

Reviewer #1: Yes

Reviewer #2: No

To clarify, we have added the following Data Availability statement to pages 15-16 of the manuscript: “Since this study is a scoping review, all data underlying the findings will be derived from published, peer-reviewed articles or grey literature that are already available in the public domain. A complete list of included studies will be provided as a supplementary file with the final manuscript. The extracted data will also be made available as supplementary tables at the time of publication.”

Reviewer #1:

Dr. Tailor and colleagues have developed a well-structured and clearly articulated review protocol aimed at summarizing how antidepressant continuation before, during, and after pregnancy is addressed in human studies. The protocol is methodologically sound and thoughtfully prepared. I have only two minor comments and questions that the authors may wish to consider:

1. Table 1. Eligibility criteria: Under the inclusion criteria, the authors state: “Peer-reviewed studies published from 2022 onward will be included in the full-text review.” In contrast, under the exclusion criteria, they write: “Peer-reviewed studies published prior to 2022 will be excluded from the full-text review.” While both statements are technically consistent, presenting the same information in both sections creates redundancy and may be confusing. Moreover, exclusion from full-text review implies that earlier studies will not be screened at that stage, not that they meet the exclusion criteria per se. I suggest rephrasing this section for clarification.

We thank the reviewer for pointing out the potential redundancy. To clarify, different publication date cutoffs are applied at different screening stages: studies published before 2015 are excluded during title/abstract screening, while studies published before 2022 are excluded at the full-text review stage. This change in cut-off time was done to improve feasibility, given the high volume of available literature, while focusing on contemporary patterns of antidepressant exposure that reflect changes in current prescribing practices. Table 1 has been revised to explicitly reflect this distinction.

2. The decision to limit the full-text review to studies published between 2022 and 2024 is under-standable, particularly given the high volume of eligible studies and the desire to reflect contempo-rary patterns of antidepressant use during pregnancy. However, to strengthen the rationale for this cutoff, the authors might consider conducting a consistency check between the title/abstract screening and full-text review stages within the 2022–2024 subset. Specifically, it would be in-formative to report on the proportion of studies that are included after full-text review but would have been missed if relying solely on title and abstract screening. This would help demonstrate the added value of the full-text review in refining study inclusion and provide empirical justification for the chosen approach.

We thank the reviewer for their helpful suggestion. As per scoping review methods, all studies that meet the title/abstract screening criteria automatically proceed to full-text review. The full-text stage is intended to apply more detailed eligibility criteria and extract relevant data, rather than to identify studies missed at the title/abstract stage. Therefore, all studies within the 2022–2024 subset that pass title/abstract screening are evaluated in full, and a “consistency check” is not applicable. This workflow is now clarified in the Methods section (page 15), as follows, “All studies identified in the title/abstract screening stage that meet the inclusion criteria will proceed to full-text review. At the full-text review stage, we will apply more detailed eligibility criteria and extract relevant data.

Reviewer #2:

Thank you for the opportunity to review this scoping review protocol. This scoping review aims to summarize how prior studies have measured antidepressant continuation/discontinuation prior to, during, and after pregnancy. While this is an important topic, I have several concerns that the authors might want to consider.

1. While the authors acknowledged that "timing of medication initiation is crucial in studies examining medication use during pregnancy" and provided examples of biological windows relevant to specific outcomes, the protocol did not attempt to classify studies by outcome type. This lack of focus led to the inclusion of a large number of studies, far exceeding the capacity of the research team to manage effectively. As a result, it seems infeasible to issue meaningful recommendations for future research based on the findings of this review. It also raised concerns about the pragmatic rationale behind including studies from 2015 based only on title and abstract screening, while including studies from 2022 based on full-text review.

We thank the reviewer for their comment. The primary aim of our scoping review is to summarize how prior studies have captured antidepressant exposure prior to, during, and after pregnancy, irrespective of specific outcomes. As such, we did not classify studies by outcome type. While we agree that etiologically relevant exposure windows may differ depending on the outcome of interest, the methodological task of defining antidepressant exposure (e.g., sustained use prior to conception, continuation or discontinuation during pregnancy, medication use patterns) is independent of the outcome being studied. In addition, because our review addresses how studies have accounted for preconception exposure as well as exposure during and after pregnancy, it is intentionally outcome-independent. Our focus is therefore on the methodological approaches used to characterize pre- and perinatal antidepressant exposure, rather than a focus on outcome-specific etiologic questions, which naturally results in a broader set of included studies. To ensure feasibility, we restricted the full-text review to studies published between 2022 and 2024, while still screening studies from 2015 onward at the title/abstract stage to capture trends over time. This approach balances comprehensiveness with the practical considerations of review workload.

2. The authors noted that "more granular definitions of real-world antidepressant exposure during pregnancy are needed, particularly since outcomes have periods of sensitivity to specific treatment exposures, meaning clinicians and patients may be likely to alter treatment regimens early in or before the perinatal period." However, the feasibility of such granular definitions largely depends on the type of data available. In many cases, particularly in studies relying on self-reported data, this level of detail is simply not attainable. Despite this, the authors did not include the type of data source among the extracted variables, which represents a significant oversight.

We thank the reviewer for highlighting the importance of capturing study designs and data sources. We have revised our data extraction tool (Table 2) to include variables such as study design, data source, and type of exposure measurement. This will allow readers to evaluate how exposure definitions vary across different study designs and data types. Table 2 has been revised to reflect this change.

Please note, the revision platform does not allow me to update the Financial Disclosure section. Please update as follows: "LT is funded by a Vanier Canada Graduate Scholarship (Funding Reference Number: CGV-192654), Canadian Institutes of Health Research Michael Smith Foreign Study Supplements Award (Funding Reference Number: FSS-203589), and an Edwin S.H. Leong Centre for Healthy Children 2025 Studentship Award. She was funded by a Network for Improving Health Systems Trainee award from the University of Toronto Leslie Dan Faculty of Pharmacy and Dalla Lana School of Public Health at the start of this work."

---

## [Editor Report · Decision Letter 1]

8 Oct 2025

Measuring the continuation of antidepressant exposure prior to, during, and after pregnancy: a scoping review protocol

PONE-D-25-32258R1

Dear Dr. Tailor,

We’re pleased to inform you that your manuscript has been judged scientifically suitable for publication and will be formally accepted for publication once it meets all outstanding technical requirements.

Kind regards,

Xiaoqin Liu

Academic Editor

PLOS ONE

Additional Editor Comments (optional):

The authors have done a great job in revision. Congratulations!

Reviewers' comments:

None

---

## [Editor Report · Acceptance letter]

PONE-D-25-32258R1

PLOS ONE

Dear Dr. Tailor,

I'm pleased to inform you that your manuscript has been deemed suitable for publication in PLOS ONE. Congratulations! Your manuscript is now being handed over to our production team.

Kind regards,

on behalf of

Dr. Xiaoqin Liu

Academic Editor

PLOS ONE